# Learning a generative model for validity in complex discrete structures

**David Janz**
University of Cambridge
dj343@cam.ac.uk

**Jos van der Westhuizen**
University of Cambridge
jv365@cam.ac.uk

**Brooks Paige**
Alan Turing Institute
University of Cambridge
bpaige@turing.ac.uk

**Matt J. Kusner**
Alan Turing Institute
University of Warwick
mkusner@turing.ac.uk

**José Miguel Hernández-Lobato**
Alan Turing Institute
University of Cambridge
jmh233@cam.ac.uk

## Abstract

Deep generative models have been successfully used to learn representations for high-dimensional discrete spaces by representing discrete objects as sequences and employing powerful sequence-based deep models. Unfortunately, these sequence-based models often produce invalid sequences: sequences which do not represent any underlying discrete structure; invalid sequences hinder the utility of such models. As a step towards solving this problem, we propose to learn a deep recurrent validator model, which can estimate whether a partial sequence can function as the beginning of a full, valid sequence. This validator provides insight as to how individual sequence elements influence the validity of the overall sequence, and can be used to constrain sequence based models to generate valid sequences – and thus faithfully model discrete objects. Our approach is inspired by reinforcement learning, where an oracle which can evaluate validity of complete sequences provides a sparse reward signal. We demonstrate its effectiveness as a generative model of Python 3 source code for mathematical expressions, and in improving the ability of a variational autoencoder trained on SMILES strings to decode valid molecular structures.

## 1 Introduction

Deep generative modeling has seen many successful recent developments, such as producing realistic images from noise (Radford et al., 2015) and creating artwork (Gatys et al., 2016). We find particularly promising the opportunity to leverage deep generative models for search in high-dimensional discrete spaces (Gómez-Bombarelli et al., 2016b; Kusner et al., 2017). Discrete search is at the heart of problems in drug discovery (Gómez-Bombarelli et al., 2016a), natural language processing (Bowman et al., 2016; Guimaraes et al., 2017), and symbolic regression (Kusner et al., 2017).

The application of deep modeling to search involves 'lifting' the search from the discrete space to a continuous space, via an autoencoder (Rumelhart et al., 1985). An autoencoder learns two mappings: 1) a mapping from discrete space to continuous space called an *encoder*; and 2) a reverse mapping from continuous space back to discrete space called a *decoder*. The discrete space is presented to the autoencoder as a sequence in some formal language — for example, in Gómez-Bombarelli et al. (2016b) molecules are encoded as SMILES strings — and powerful sequential models (e.g., LSTMs (Hochreiter & Schmidhuber, 1997) GRUs (Cho et al., 2014), DCNNs (Kalchbrenner et al., 2014)) are applied to the string representation. When employing these models as encoders and decoders, generation of invalid sequences is however possible, and using current techniques this happens frequently. Kusner et al. (2017) aimed to fix this by basing the sequential models on parse tree representations of the discrete structures, where externally specified grammatical rules assist the model in the decoding process. This work boosted the ability of the model to produce valid sequences

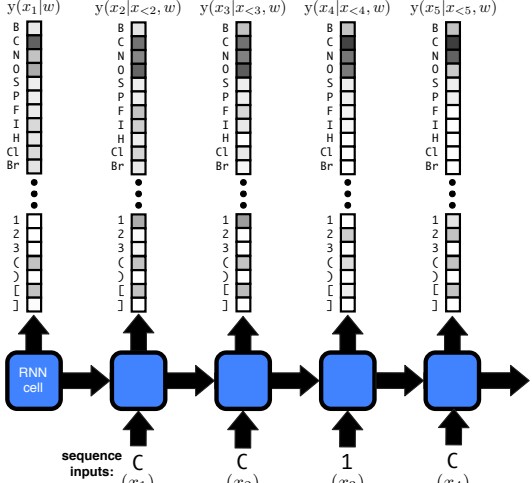

Figure 1: The recurrent model used to approximate the $Q$-function. A hypothetical logistic function activation is shown for each character in $\mathcal{C}$. Here the set of characters is the SMILES alphabet and we use the first 3 characters of the molecule in figure 3 as the input example. The initial character is predicted from the first hidden state, and the LSTM continues until the end of the sequence.

during decoding, but its performance achieved by this method leaves scope for improvement, and the method requires hand-crafted grammatical rules for each application domain.

In this paper, we propose a generative approach to modeling validity that can learn the validity constraints of a given discrete space. We show how concepts from reinforcement learning may be used to define a suitable generative model and how this model can be approximated using sequence-based deep learning techniques. To assist in training this generative model we propose two data augmentation techniques. Where no labeled data set of valid and invalid sequences is available, we propose a novel approach to active learning for sequential tasks inspired by classic mutual-information-based approaches (Houlsby et al., 2011; Hernández-Lobato et al., 2014). In the context of molecules, where data sets containing valid molecule examples do exist, we propose an effective data augmentation process based on applying minimal perturbations to known-valid sequences. These two techniques allow us to rapidly learn sequence validity models that can be used as a) generative models, which we demonstrate in the context of Python 3 mathematical expressions and b) a grammar model for character-based sequences, that can drastically improve the ability of deep models to decode valid discrete structures from continuous representations. We demonstrate the latter in the context of molecules represented as SMILES strings.

## 2 A MODEL FOR SEQUENCE VALIDITY

To formalise the problem we denote the set of discrete sequences of length $T$ by $\mathcal{X} = \{(x_1, \ldots, x_T) : x_t \in \mathcal{C}\}$ using an alphabet $\mathcal{C} = \{1, \ldots, C\}$ of size $C$. Individual sequences in $\mathcal{X}$ are denoted $x_{1:T}$. We assume the availability of a *validator* $v \colon \mathcal{X} \to \{0, 1\}$, an oracle which can tell us whether a given sequence is valid. It is important to note that such a validator gives very sparse feedback: it can only be evaluated on a *complete* sequence. Examples of such validators are compilers for programming languages (which can identify syntax and type errors) and chemo-informatics software for parsing SMILES strings (which identify violations of valence constraints). Running the standard validity checker $v(x_{1:T})$ on a partial sequence or subsequence (e.g., the first $t < T$ characters of a computer program) does not in general provide any indication as to whether the complete sequence of length $T$ is valid.

We aim to obtain a generative model for the sequence set $\mathcal{X}_+ = \{x_{1:T} \in \mathcal{X} : v(x_{1:T}) = 1\}$, the subset of valid sequences in $\mathcal{X}$. To achieve this, we would ideally like to be able to query a more-informative function $\tilde{v}(x_{1:t})$ which operates on prefixes $x_{1:t}$ of a hypothetical longer sequence $x_{1:T}$ and outputs

$$\tilde{v}(x_{1:t}) = \begin{cases} 1 & \text{if there exists a suffix } x_{t+1:T} \text{ such that } v([x_{1:t}, x_{t+1:T}]) = 1, \\ 0 & \text{otherwise} \end{cases} \tag{1}$$

where $[x_{1:t}, x_{t+1:T}]$ concatenates a prefix and a suffix to form a complete sequence. The function $\tilde{v}(x_{1:t})$ can be used to determine whether a given prefix can ever successfully yield a valid outcome.

Note that we are indifferent to *how many* suffixes yield valid sequences. With access to $\tilde{v}(x_{1:t})$, we could create a generative model for $\mathcal{X}_+$ which constructs sequences from left to right, a single character at a time, using $\tilde{v}(x_{1:t})$ to provide early feedback as to which of the next character choices will surely not lead to a "dead end" from which no valid sequence can be produced.

We frame the problem of modeling $\mathcal{X}_+$ as a Markov decision process (Sutton & Barto, 1998) for which we train a reinforcement learning agent to select characters sequentially in a manner that avoids producing invalid sequences. At time $t = 1, \ldots, T$, the agent is in state $x_{<t} = x_{1:t-1}$ and can take actions $x_t \in \mathcal{C}$. At the end of an episode, following action $x_T$, the agent receives a reward of $v(x_{1:T})$. Since in practice we are only able to evaluate $v(x_{1:T})$ in a meaningful way on complete sequences, the agent does not receive any reward at any of the intermediate steps $t < T$. The optimal $Q$-function $Q^\star(s, a)$ (Watkins, 1989), a function of a state $s$ and an action $a$, represents the expected reward of an agent following an optimal policy which takes action $a$ at state $s$. This optimal $Q$-function assigns value 1 to actions $a = x_t$ in state $s = x_{<t}$ for which there exists a suffix $x_{t+1:T}$ such that $[x_{1:t}, x_{t+1:T}] \in \mathcal{X}_+$, and value 0 to all other state/action pairs. This behaviour exactly matches the desired prefix validator in (1), that is, $Q^\star(x_{<t}, x_t) = \tilde{v}(x_{1:t})$, and so for the reinforcement learning environment as specified, learning $\tilde{v}(x_{1:t})$ corresponds to learning the $Q$-function.

Having access to $Q^\star$ would allow us to obtain a generative model for $\mathcal{X}_+$. In particular, an agent following any optimal policy $\pi^\star(x_{<t}) = \mathrm{argmax}_{x_t \in \mathcal{C}} Q^\star(x_{<t}, x_t)$ will always generate valid sequences. If we sample uniformly at random across all optimal actions at each time $t = 1, \ldots, T$, we obtain the joint distribution given by

$$p(x_{1:T}) = \prod_{t=1}^{T} \frac{Q^\star(x_{<t}, x_t)}{Z(x_{<t})}, \tag{2}$$

where $Z(x_{<t}) = \sum_{x_t} Q^\star(x_{<t}, x_t)$ are the per-timestep normalisation constants. This distribution allows us to sample sequences $x_{1:T}$ in a straightforward manner by sequentially selecting characters $x_t \in \mathcal{C}$ given the previously selected ones in $x_{<t}$.

In this work we focus on learning an approximation to (2). For this, we use recurrent neural networks, which have recently shown remarkable empirical success in the modeling of sequential data, e.g., in natural language processing applications (Sutskever et al., 2014). We approximate the optimal $Q$-function with a long-short term memory (LSTM) model (Hochreiter & Schmidhuber, 1997) that has one output unit per character in $\mathcal{C}$, with each output unit using a logistic activation function (see figure 1), such that the output is in the closed interval $[0, 1]$. We denote by $\mathrm{y}(x_t|x_{<t}, w)$ the value at time $t$ of the LSTM output unit corresponding to character $x_t$ when the network weights are $w$ the input is the sequence $x_{<t}$. We interpret the neural network output $\mathrm{y}(x_t|x_{<t}, w)$ as $p(Q^\star(x_{<t}, x_t) = 1)$, that is, as the probability that action $x_t$ can yield a valid sequence given that the current state is $x_{<t}$.

Within our framing a sequence $x_{1:T}$ will be valid according to our model if every action during the sequence generation process is permissible, that is, if $Q^\star(x_{<t}, x_t) = 1$ for $t = 1, \ldots, T$. Similarly, we consider that the sequence $x_{1:T}$ will be invalid if at least one action during the sequence generation process is not valid[1], that is, if $Q^\star(x_{<t}, x_t) = 0$ at least once for $t = 1, \ldots, T$. This specifies the following log-likelihood function given a training set $\mathcal{D} = \{(x_{1:T}^n, y_n)\}_{n=1}^{N}$ of sequences $x_{1:T}^n \in \mathcal{X}$ and corresponding labels $y_n = v(x_{1:T})$:

$$\mathcal{L}(w|\mathcal{D}) = \sum_{n=1}^{N} \{y_n \log p(y_n = 1|x_{1:T}^n, w) + (1 - y_n) \log p(y_n = 0|x_{1:T}^n, w)\}, \tag{3}$$

where, following from the above characterisation of valid and invalid sequences, we define

$$p(y_n = 1|x_{1:T}^n, w) = \prod_{t=1}^{T} \mathrm{y}(x_t|x_{<t}, w), \qquad p(y_n = 0|x_{1:T}^n, w) = 1 - \prod_{t=1}^{T} \mathrm{y}(x_t|x_{<t}, w), \tag{4}$$

according to our model's predictions. The log-likelihood (3) can be optimised using backpropagation and stochastic gradient descent and, given sufficient model capacity, results in a maximiser $\hat{w}$ such that $\mathrm{y}(x_t|x_{<t}, \hat{w}) \approx Q^\star(x_{<t}, x_t)$.

---

[1] Note that, once $Q^\star(x_{<t}, x_t)$ is zero, all the following values of $Q^\star(x_{<t}, x_t)$ in that sequence will be irrelevant to us. Therefore, we can safely assume that a sequence is invalid if $Q^\star(x_{<t}, x_t)$ is zero at least once in the sequence.

Instead of directly maximising (3), we can follow a Bayesian approach to obtain estimates of uncertainty in the predictions of our LSTM model. For this, we can introduce dropout layers which stochastically zero-out units in the input and hidden layers of the LSTM model according to a Bernoulli distribution (Gal & Ghahramani, 2016). Under the assumption of a Gaussian prior $p(w)$ over weights, the resulting stochastic process yields an implicit approximation $q(w)$ to the posterior distribution $p(w|\mathcal{D}) \propto \exp(\mathcal{L}(w|\mathcal{D}))p(w)$. We do this to obtain uncertainty estimates, allowing us to perform efficient active learning, as described in section 3.1.

## 3 ONLINE GENERATION OF SYNTHETIC TRAINING DATA

One critical aspect of learning $w$ as described above is how to generate the training set $\mathcal{D}$ in a sensible manner. A naïve approach could be to draw elements from $\mathcal{X}$ uniformly at random. However, in many cases, $\mathcal{X}$ contains only a tiny fraction of valid sequences and the uniform sampling approach produces extremely unbalanced sets which contain very little information about the structure of valid sequences. While rejection sampling can be used to increase the number of positive samples, the resulting additional cost makes such an alternative infeasible in most practical cases. The problem gets worse as the length of the sequences considered $T$ increases since $|\mathcal{X}|$ will always grow as $|\mathcal{C}|^T$, while $|\mathcal{X}_+|$ will typically grow at a lower rate.

We employ two approaches for artificially constructing balanced sets that permit learning these models in far fewer samples than $|\mathcal{C}|^T$. In settings where we do not have a corpus of known valid sequences, Bayesian active learning can automatically construct the training set $\mathcal{D}$. This method works by iteratively selecting sequences in $\mathcal{X}$ that are maximally informative about the model parameters $w$ given the data collected so far (MacKay, 1992). When we do have a set of known valid sequences, we use these to seed a process for generating balanced sets by applying random perturbations to valid sequences.

### 3.1 ACTIVE LEARNING

Let $x_{1:T}$ denote an arbitrary sequence and let $y$ be the unknown binary label indicating whether $x_{1:T}$ is valid or not. Our model's predictive distribution for $y$, that is, $p(y|x_{1:T}, w)$ is given by (4). The amount of information on $w$ that we expect to gain by labeling and adding $x_{1:T}$ to $\mathcal{D}$ can be measured in terms of the expected reduction in the entropy of the posterior distribution $p(w|\mathcal{D})$. That is,

$$\alpha(x_{1:T}) = \mathrm{H}[p(w|\mathcal{D})] - \mathbb{E}_{p(y|x_{1:T}, w)}\mathrm{H}[p(w|\mathcal{D} \cup (x_{1:T}, y)]\,, \qquad (5)$$

where $\mathrm{H}(\cdot)$ computes the entropy of a distribution. This formulation of the entropy-based active learning criterion is, however, difficult to approximate, because it requires us to condition on $x_{1:T}$ – effectively . To obtain a simpler expression we follow Houlsby et al. (2011) and note that $\alpha(x_{1:T})$ is equal to the mutual information between $y$ and $w$ given $x_{1:T}$ and $\mathcal{D}$

$$\alpha(x_{1:T}) = \mathrm{H}\{\mathbb{E}_{p(w|\mathcal{D})}[p(y|x_{1:T}, w)]\} - \mathbb{E}_{p(w|\mathcal{D})}\{\mathrm{H}[p(y|x_{1:T}, w)]\}\,, \qquad (6)$$

which is easier to work with as the required entropy is now that of Bernoulli predictive distributions, an analytic quantity. Let $\mathcal{B}(p)$ denote a Bernoulli distribution with probability $p$, and with probability mass $p^z(1-p)^{1-z}$ for values $z \in \{0, 1\}$. The entropy of $\mathcal{B}(p)$ can be easily obtained as

$$\mathrm{H}[\mathcal{B}(p)] = -p \log p - (1-p) \log(1-p) \equiv g(p)\,. \qquad (7)$$

The expectation with respect to $p(w|\mathcal{D})$ can be easily approximated by Monte Carlo. We could attempt to sequentially construct $\mathcal{D}$ by optimising (6). However, this optimisation process would still be difficult, as it would require evaluating $\alpha(x_{1:T})$ exhaustively on all the elements of $\mathcal{X}$. To avoid this, we follow a greedy approach and construct our informative sequence in a sequential manner. In particular, at each time step $t = 1, \ldots, T$, we select $x_t$ by optimising the mutual information between $w$ and $Q^\star(x_{<t}, x_t)$, where $x_{<t}$ denotes here the prefix already selected at previous steps of the optimisation process. This mutual information quantity is denoted by $\alpha(x_t|x_{<t})$ and its expression is given by

$$\alpha(x_t|x_{<t}) = \mathrm{H}\{\mathbb{E}_{p(w|\mathcal{D})}[\mathcal{B}(\mathrm{y}(x_t|x_{<t}, w))]\} - \mathbb{E}_{p(w|\mathcal{D})}\{\mathrm{H}[\mathcal{B}(\mathrm{y}(x_t|x_{<t}, w))]\}\,. \qquad (8)$$

The generation of an informative sequence can then be performed efficiently by sequentially optimising (8), an operation that requires only $|\mathcal{C}| \times T$ evaluations of $\alpha(x_t|x_{<t})$.

To obtain an approximation to (8), we first approximate the posterior distribution $p(w|\mathcal{D})$ with $q(w)$ and then estimate the expectations in (8) by Monte Carlo using $K$ samples drawn from $q(w)$. The resulting estimator is given by

$$\hat{\alpha}(x_t \mid x_{<t}) = g\left[\frac{1}{K}\sum_{k=1}^{K}\mathrm{y}(x_t|x_{<t},w_k)\right] - \frac{1}{K}\sum_{k=1}^{K}g\left[\mathrm{y}(x_t|x_{<t},w_k)\right], \tag{9}$$

where $w_1,\ldots,w_K \sim q(w)$ and $g(\cdot)$ is defined in (7). The nonlinearity of $g(\cdot)$ means that our Monte Carlo approximation is biased, but still consistent. We found that reasonable estimates can be obtained even for small $K$. In our experiments we use $K = 16$.

The iterative procedure just described is designed to produce a single informative sequence. In practice, we would like to generate a batch of informative and diverse sequences. The reason for this is that, when training neural networks, processing a batch of data is computationally more efficient than individually processing multiple data points. To construct a batch with $L$ informative sequences, we propose to repeat the previous iterative procedure $L$ times. To introduce diversity in the batch-generation process, we "soften" the greedy maximisation operation at each step by injecting a small amount of noise in the evaluation of the objective function (Finkel et al., 2006). Besides introducing diversity, this can also lead to better overall solutions than those produced by the noiseless greedy approach (Cho, 2016). We introduce noise into the greedy selection process by sampling from

$$p(x_t|x_{<t},\theta) = \frac{\exp\{\alpha(x_t|x_{<t})/\theta\}}{\sum_{x'_t \in \mathcal{C}}\exp\{\alpha(x'_t|x_{<t})/\theta\}} \tag{10}$$

for each $t = 1,\ldots,T$, which is a Boltzmann distribution with sampling temperature $\theta$. By adjusting this temperature parameter, we can trade off the diversity of samples in the batch vs. their similarity.

## 3.2 Data augmentation

In some settings, such as the molecule domain we will consider later, we have databases of known-valid examples (e.g. collections of known drug-like molecules), but rarely are sets of invalid examples available. Obtaining invalid sequences may seem trivial, as invalid samples may be obtained by sampling uniformly from $\mathcal{X}$, however these are almost always so far from any valid sequence that they carry little information about the boundary of valid and invalid sequences. Using just a known data set also carries the danger of overfitting to the subset of $\mathcal{X}_+$ covered by the data.

We address this by perturbing sequences from a database of valid sequences, such that approximately half of the thus generated sequences are invalid. These perturbed sequences $x'_{1:T}$ are constructed by setting each $x'_t$ to be a symbol selected independently from $\mathcal{C}$ with probability $\gamma$, while remaining the original $x_t$ with probability $1 - \gamma$. In expectation this changes $\gamma T$ entries in the sequence. We choose $\gamma = 0.05$, which results in synthetic data that is approximately 50% valid.

## 4 Experiments

We test the proposed validity checker in two environments. First, we look at fixed length Python 3 mathematical expressions, where we derive lower bounds for the support of our model and compare the performance of active learning with that achieved by a simple passive approach. Secondly, we look at molecular structures encoded into string representation, where we utilise existing molecule data sets together with our proposed data augmentation method to learn the rules governing molecule string validity. We test the efficacy of our validity checker on the downstream task of decoding valid molecules from a continuous latent representation given by a variational autoencoder.

## 4.1 Mathematical expressions

We illustrate the utility of the proposed validity model and sequential Bayesian active learning in the context of Python 3 mathematical expressions. Here, $\mathcal{X}$ consists of all length 25 sequences that can be constructed from the alphabet of numbers and symbols shown in table 1. The validity of any given expression is determined using the Python 3 `eval` function: a valid expression is one that does not raise an exception when evaluated.

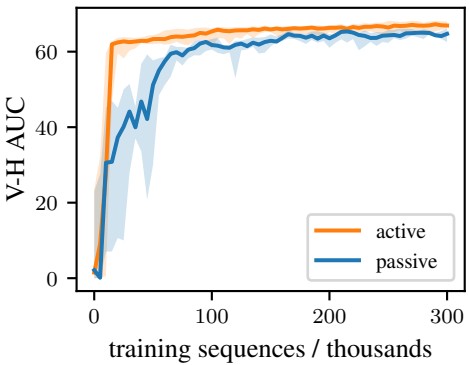 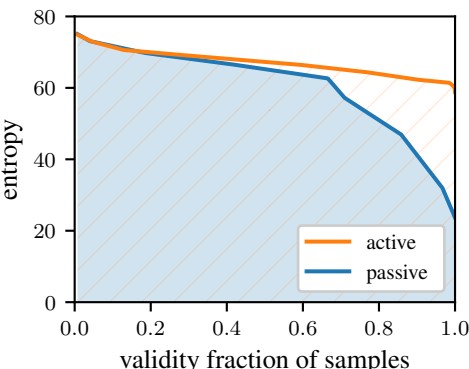

Figure 2: Experiments with length 25 Python expressions. (Left) Area under validity-entropy curve as training progresses, 10-90 percentiles shaded. Active learning converges faster and reaches a higher maximum. (Right) Entropy versus validity for median active and median passive model after 200k training sequences. Both models have learnt a high entropy distribution over valid sequences.

**Measuring model performance**  Within this problem domain we do not assume the existence of a data set of positive examples. Without a validation data set to measure performance on, we compare the models in terms of their capability to provide high entropy distributions over valid sequences. We define a generative procedure to sample from the model and measure the validity and entropy of the samples. To sample stochastically, we use a Boltzmann policy, i.e. a policy which samples next actions according to

$$\pi(x_t = c|x_{<t}, w, \tau) = \frac{\exp(\mathrm{y}(c|x_{<t}, w)/\tau)}{\sum_{j \in \mathcal{C}} \exp(\mathrm{y}(j|x_{<t}, w)/\tau)} \tag{11}$$

where $\tau$ is a temperature constant that governs the trade-off between exploration and exploitation. Note that this is not the same as the Boltzmann distribution used as a proposal generation scheme during active learning, which was defined not on $Q$-function values but rather on the estimated mutual information.

We obtain samples $\{x^{(1)}, \ldots, x^{(N)}\}_{\tau_i}$ for a range of temperatures $\tau_i$ and compute the validity fraction and entropy of each set of samples. These points now plot a curve of the trade-off between validity and entropy that a given model provides. Without a preferred level of sequence validity, the area under this validity-entropy curve (V-H AUC) can be utilised as a metric of model quality. To provide some context for the entropy values, we estimate an information theoretic lower bound for the fraction of the set $\mathcal{X}_+$ that our model is able to generate. This translates to upper bounding the false negative rate for our model.

**Experimental setup and results**  We train two models using our proposed Q-function method: *passive*, where training sequences are sampled from a uniform distribution over $\mathcal{X}$, and *active*, where we use the procedure described in section 3.1 to select training sequences. The two models are otherwise identical.

Both trained models give a diverse output distribution over valid sequences (figure 2). However, as expected, we find that the *active* method is able to learn a model of sequence validity much more rapidly than sampling uniformly from $\mathcal{X}$, and the corresponding converged model is capable of generating many more distinct valid sequences than that trained using the *passive* method. In table 2

Table 1: Python 3 expression alphabet

| digits | operators | comparisons | brackets |
|---|---|---|---|
| 1234567890 | +-*/%! | =<> | () |

Table 2: Estimated lower bound of coverage $N$ for *passive* and *active* models, defined as the size of the set of Python expressions on which the respective model places positive probability mass. Evaluation is on models trained until convergence ($800,000$ training points, beyond the scope of figure 2). The lower bound estimation method is detailed in Appendix A.

| temperature $\tau$ | passive model | | active model | |
|---|---|---|---|---|
| | validity | $N$ | validity | $N$ |
| 0.100 | 0.850 | $9.7 \times 10^{27}$ | 0.841 | $8.2 \times 10^{28}$ |
| 0.025 | 0.969 | $2.9 \times 10^{25}$ | 0.995 | $4.3 \times 10^{27}$ |
| 0.005 | 1.000 | $1.1 \times 10^{22}$ | 1.000 | $1.3 \times 10^{27}$ |

we present lower bounds on the support of the two respective models. The details of how this lower bound is computed can be found in appendix A. Note that the overhead of the active learning data generating procedure is minimal: processing 10,000 takes 31s with *passive* versus 37s with *active*.

## 4.2 SMILES MOLECULES

SMILES strings (Weininger, 1970) are one of the most common representations for molecules, consisting of an ordering of atoms and bonds. It is attractive for many applications because it maps the graphical representation of a molecule to a sequential representation, capturing not just its chemical composition but also structure. This structural information is captured by intricate dependencies in SMILES strings based on chemical properties of individual atoms and valid atom connectivities. For instance, the atom Bromine can only bond with a single other atom, meaning that it may only occur at the beginning or end of a SMILES string, or within a so-called 'branch', denoted by a bracketed expression `(Br)`. We illustrate some of these rules, including a Bromine branch, in figure 3, with a graphical representation of a molecule alongside its corresponding SMILES string. There, we also show examples of how a string may fail to form a valid SMILES molecule representation. The full SMILES alphabet is presented in table 3.

Table 3: SMILES alphabet

| atoms/chirality | bonds/ringbonds | charges | branches/brackets |
|---|---|---|---|
| `B C N O S P F I H Cl Br @` | `= # / \ 1 2 3 4 5 6 7 8` | `- +` | `( ) [ ]` |

The intricacy of SMILES strings makes them a suitable testing ground for our method. There are two technical distinctions to make between this experimental setup and the previously considered Python 3 mathematical expressions. As there exist databases of SMILES strings, we leverage those by using the data augmentation technique described in section 3.2. The main data source considered is the ZINC data set Irwin & Shoichet (2005), as used in Kusner et al. (2017). We also use the USPTO 15k reaction products data (Lowe, 2014) and a set of molecule solubility information (Huuskonen, 2000) as withheld test data. Secondly, whereas we used fixed length Python 3 expressions in order to obtain coverage bounds, molecules are inherently of variable length. We deal with this by padding all molecules to fixed length.

**Validating grammar model accuracy** As a first test of the suitability of our proposed validity model, we train it on augmented ZINC data and examine the accuracy of its predictions on a withheld test partition of that same data set as well as the two unseen molecule data sets. Accuracy is the ability of the model to accurately recognise which perturbations make a certain SMILES string invalid, and which leave it valid – effectively how well the model has captured the grammar of SMILES strings in the vicinity of the data manifold. Recalling that a sequence is invalid if $\tilde{v}(x_{1:t}) = 0$ at any $t \leq T$, we consider the model prediction for molecule $x_{1:T}$ to be $\prod_{t=1}^{T} \mathbb{I}\left[ y(x_t | x_{<t}, w) \geq 0.5 \right]$, and compare this to its true label as given by rdkit, a chemical informatics software. The results are encouraging, with the model achieving 0.998 accuracy on perturbed ZINC (test) and 1.000 accuracy on both perturbed USPTO and perturbed Solubility withheld data. Perturbation rate was selected such that approximately half of the perturbed strings are valid.

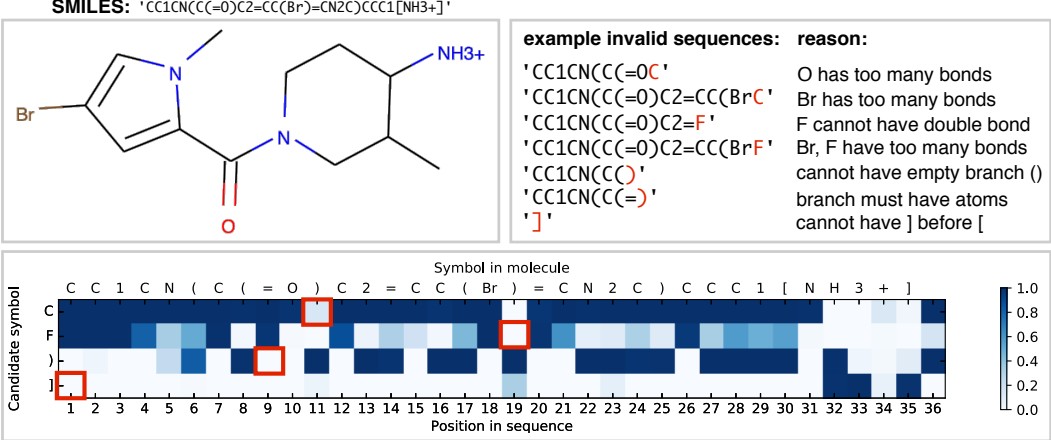

Figure 3: Predictions $y(x_t|x_{<t}, w)$ of the agent at each step $t$ for the valid test molecule shown in the top left figure, for a subset of possible actions (selecting as next character C, F, ), or ]). Each column shows which actions the trained agent believes are valid at each $t$, given the characters $x_{<t}$ preceding it. We see that the validity model has learned basic valence constraints: for example the oxygen atom O at position 10 can form at most 2 bonds, and since it is preceded by a double bond, the model knows that neither carbon C nor fluorine F can immediately follow it at position 11; we see the same after the bromine Br at position 18, which can only form a single bond. The model also correctly identifies that closing branch symbols ) cannot immediately follow opening branches (after positions 6, 8, and 17), as well as that closing brackets ] cannot occur until an open bracket has been followed by at least one atom (at positions 32–35). The full output heatmap for this example molecule is shown in Figure 4 in the appendix.

**Integrating with Character VAE**    To demonstrate the models capability of improving preexisting generative models for discrete structures, we show how it can be used to improve the results of previous work, a character variational autoencoder (CVAE) applied to SMILES strings (Gómez-Bombarelli et al., 2016b; Kingma & Welling, 2013). Therein, an encoder maps points in $\mathcal{X}_+$ to a continuous latent representation $\mathcal{Z}$ and a paired decoder maps points in $\mathcal{Z}$ back to $\mathcal{X}_+$. A reconstruction based loss is minimised such that training points mapped to the latent space decode back into the same SMILES strings. The fraction of test points that do is termed reconstruction accuracy. The loss also features a term that encourages the posterior over $\mathcal{Z}$ to be close to some prior, typically a normal distribution. A key metric for the performance of variational autoencoder models for discrete structures is the fraction of points sampled from the prior over $\mathcal{Z}$ that decode into valid molecules. If many points do not correspond to valid molecules, any sort of predictive modeling on that space will likely also mostly output invalid SMILES strings.

The decoder functions by outputting a set of weights $f(x_t|z)$ for each character $x_t$ in the reconstructed sequence conditioned on a latent point $z \in \mathcal{Z}$; the sequence is recovered by sampling from a multinomial according to these weights. To integrate our validity model into this framework, we take the decoder output for each step $t$ and mask out choices that we predict cannot give valid sequence continuations. We thus sample characters with weights given by $f(x_t|z) \cdot \mathbb{I}\left[y(x_t|x_{<t}, w) \geq 0.5\right]$.

**Autoencoding benchmarks**    Table 4 contains a comparison of our work to a plain CVAE and to the Grammar VAE approach. We use a Kekulé format of the ZINC data in our experiments, a specific representation of aromatic bonds that our model handled particularly well. Note that the results we quote for Grammar VAE are taken directly from Kusner et al. (2017) and on non-Kekulé format data. The CVAE model is trained for 100 epochs, as per previous work – further training improves reconstruction accuracy.

We note that the binary nature of the proposed grammar model means that it does not affect the reconstruction accuracy. In fact, some modest gains are present. The addition of our grammar model to the character VAE significantly improves its ability to decode discrete structures, as seen by the order of magnitude increase in latent sample validity. The action of our model is completely post-hoc

and thus can be applied to any pre-trained character-based VAE model where elements of the latent space correspond to a structured discrete sequence.

| Model | reconstruction accuracy | sample validity |
|---|---|---|
| CVAE + Validity Model | 50.2% | 22.3% |
| Grammar VAE | 53.7% | 7.2% |
| Plain CVAE | 49.7% | 0.5% |

Table 4: Performance metrics for VAE-based molecule model trained for 100 epochs on ZINC (train) data, with and without the proposed validity model overlaid at test time, and the Grammar VAE method. Sample validity is the fraction of samples from the prior over $\mathcal{Z}$ that decode into valid molecules.

## 5 DISCUSSION

In this work we proposed a modeling technique for learning the validity constraints of discrete spaces. The proposed likelihood makes the model easy to train, is unaffected by the introduction of padding for variable length sequences and, as its optimum is largely independent of the training data distribution, it allows for the utilisation of active learning techniques. Through experiments we found that it is vital to show the model informative examples of validity constraints being validated. Thus, where no informative data sets exist, we proposed a mutual-information-based active learning scheme that uses model uncertainty to select training sequences. We used principled approximations to make that learning scheme computationally feasible. Where data sets of positive examples are available, we proposed a simple method of perturbations to create informative examples of validity constraints being broken.

The model showed promise on the Python mathematical expressions problem, especially when combined with active learning. In the context of SMILES molecules, the model was able to learn near-perfectly the validity of independently perturbed molecules. When applied to the variational autoencoder benchmark on SMILES strings, the proposed method beat the previous results by a large margin on prior sample validity – the relevant metric for the downstream utility of the latent space. The model is simple to apply to existing character-based models and is easy to train using data produced through our augmentation method. The perturbations used do not, however, capture every way in which a molecule may be mis-constructed. Correlated changes such as the insertion of matching brackets into expressions are missing from our scheme. Applying the model to a more structured representation of molecules, for example, sequences of parse rules as used in the Grammar VAE, and performing perturbations in that structured space is likely to deliver even greater improvements in performance.

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

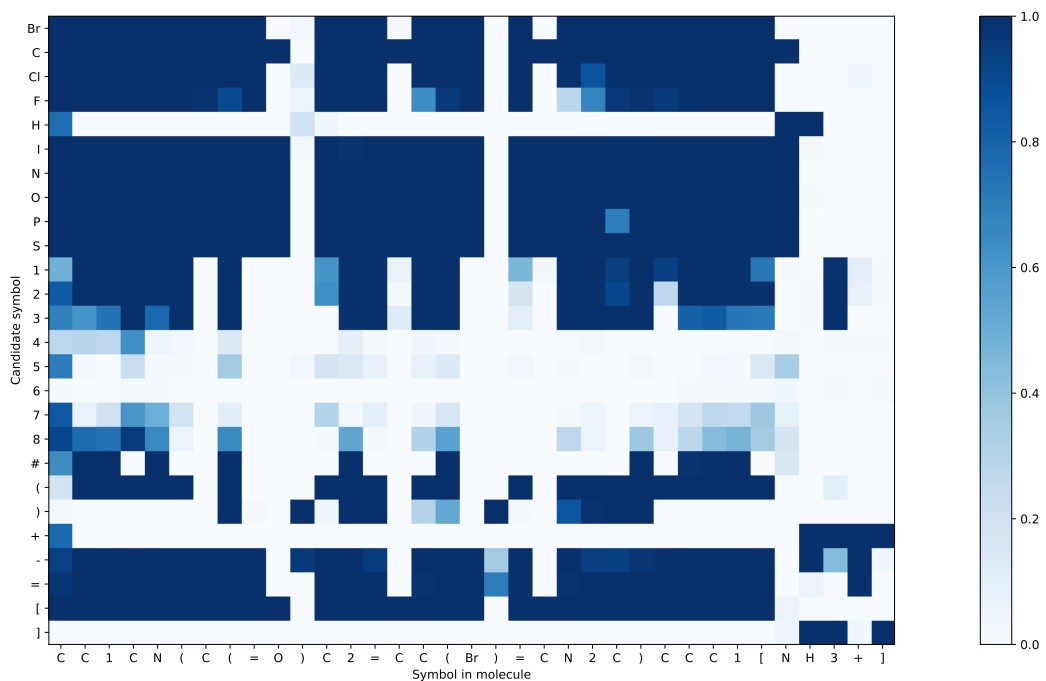

Figure 4: Full heatmap showing predictions $y(x_t|x_{<t}, w)$ for the molecule in Figure 3.

## A   COVERAGE ESTIMATION

Ideally, we would like to check that the learned model $y(x_t|x_{<t}, w)$ assigns positive probability to exactly those points which may lead to valid sequences, but for large discrete spaces this is impossible to compute or even accurately estimate. A simple check for accuracy could be to evaluate whether the model correctly identifies points as valid in a known, held-out validation or test set of real data, relative to randomly sampled sequences (which are nearly always invalid). However, if the validation set is too "similar" to the training data, even showing 100% accuracy in classifying these as valid may simply indicate having overfit to the training data: a discriminator which identifies data as similar to the training data needs to be accurate over a much smaller space than a discriminator which estimates validity over all of $\mathcal{X}$.

Instead, we propose to evaluate the trade-off between accuracy on a validation set, and an approximation to the size of the effective support of $\prod_t y(x_t|x_{<t}, w)$ over $\mathcal{X}$. Let $\mathcal{X}_+$ denote the valid subset of $\mathcal{X}$. Suppose we estimate the valid fraction $f_+ = |\mathcal{X}_+|/|\mathcal{X}|$ by simple Monte Carlo, sampling uniformly from $\mathcal{X}$. We can then estimate $N_+ = |\mathcal{X}_+|$ by $f_+|\mathcal{X}|$, where $|\mathcal{X}| = C^T$, a known quantity. A uniform distribution over $N_+$ sequences would have an entropy of $\log N_+$. We denote the entropy of output from the model $H$. If our model was perfectly uniform over the sequences it can generate, it would then be capable of generating $N_{\text{model}} = e^H$. As our model at its optimum is extremely not uniform over sequences $x \in \mathcal{X}$, this is very much a lower bound a coverage.

