# OpenReview forum: "Learning a Generative Model for Validity in Complex Discrete Structures"
_ICLR.cc/2018/Conference — Accept (Poster)_

### Official Review · AnonReviewer2 · 2017-11-23
**Interesting paper, raises as many questions as it answers.**

**Rating:** 7
**Confidence:** 3

**Review:**

The authors use a recurrent neural network to build generative models of sequences in domains where the vast majority of sequences is invalid. The basic idea, outlined in Eq. 2, is moderately straightforward: at each step, use an approximation of the Q function for subsequences of the appropriate length to pick a valid extension. There are numerous details to get right. The writing is mostly clear, and the examples are moderately convincing. I wish the paper had more detailed arguments and discussions.

I question the appropriateness of Eq. 2 as a target. A correctly learned model will put positive weight on valid sequences, but it may be an arbitrarily slow way to generate diverse sequences, depending on the domain. For instance, imagine a domain of binary strings where the valid sequences are the all 1 sequence, or any sequence beginning with a 0. Half the generated sequences would be all 1's in this situation, right? And it's easy to construct further examples that are much worse than this?

The use of Bayesian active learning to generate the training set feels like an elegant idea. However, I wish there were more clarity about what was ad hoc and what wasn't. For instance, I think the use of  dropout to get q is suspect (see for instance https://arxiv.org/abs/1711.02989), and I'd prefer a little more detail on statements like "The nonlinearity of g(·) means that our Monte
Carlo approximation is biased, but still consistent." Do we have any way of quantifying the bias? Is the statement about K=16 being reasonable a statement about bias, variance, or both?

For Python strings:
- Should we view the fact that high values of tau give a validity of 1.0 as indicative that the domain's constraints are fairly easy to learn?
- "The use of a Boltzmann policy allows us to tune the temperature parameter to identify policies
which hit high levels of accuracy for any learned Q-function approximation." This is only true to the extent the domain is sufficiently "easy" right? Is the argument that even in very hard domains, you might get this by just having an RNN which memorized a single valid sequence (assuming at least one could be found)?
- What's the best explanation for *why* the active model has much higher diversity? I understand that the active model is picking examples that tell us more about the uncertainty in w, but it's not obvious to me that means higher diversity. Do we think this is a universal property of domains?
- The highest temperature active model is exploring about half of valid sequences (modulo the non-tightness of the bound)? Have you tried gaining some insight by generating thousands of valid sequences manually and seeing which ones the model is rejecting?
- The coverage bound is used only for for Python expressions, right? Why not just randomly sample a few thousand positives and use that to get a better estimate of coverage? Since you can sample from the true positive set, it seems that your argument from the appendix about the validation set being "too similar to the training set" doesn't apply?
- It would be better to see a comparison to a strong non-NN baseline. For instance, I could easily make a PCFG over Python math expressions, and use rejection sampling to get rid of those that aren't exactly length 25, etc.?

I question how easy the Python strings example is. In particular, it might be that it's quite an easy example (compared to the SMILES) example. For SMILES, it seems like the Bayesian active learning technique is not by itself sufficient to create a good model? It is interesting that in the solubility domain the active model outperforms, but it would be nice to see more discussion / explanation.

Minor note: The incidence of valid strings in the Python expressions domain is (I believe) > 1/5000, although I guess 1 in 10,000 is still the right order of magnitude.

If I could score between "marginal accept" and "accept" I would.

---

> ### Author Response · Authors · 2018-01-05
> **Regarding the worse performance of active learning with SMILES**
>
> In our new SMILES experiments, instead of the active learning we propose a data augmentation strategy which generates informative negative samples. Table 4 shows that this strategy allows us to outperform previous state-of-the-art results [Kusner et al. (2017)].
>
> The reason for not using our active learning strategy with SMILES is because, while it does learn to discover strings that are technically valid, they are not chemically-realistic. This is why our initial active learning results in the SMILES domain were mixed. By instead augmenting an existing set of realistic molecules, we are able to more efficiently explore the space of realistic SMILES strings.

---

> ### Author Response · Authors · 2018-01-05
> **Regarding Experiments**
>
> We have updated and extended our SMILES experiments. We now provide comparisons of our work with a state-of-art context-free grammar based approach [Kusner et al. (2017)].
>
> For python expressions, the active model sees a lot more valid sequences during training, and thus gets better at modelling a large range of those. The passive one doesn’t see as many examples of valid sequences, and so doesn’t learn their general properties as well. Looking at the generated data, both methods struggle with correlated changes like brackets. One possible fix is to use variable length sequences to learn the usage of brackets from shorter sequences, which can then be generalised to longer sequences.
>
> About the claim regarding Boltzmann sampling being used to generate high validity samples at low enough temperatures, indeed, we mean that it can just generate the same one valid sequence with no sequence diversity. Tau at 1.0 validity could perhaps give an indication of the difficulty of the problem domain.

---

> ### Author Response · Authors · 2018-01-05
> **Regarding the bias in the Monte Carlo approximation**
>
> We have investigated the quality of the biased Monte Carlo information gain estimator. For active learning, the bias would only matter if it affects the relative ordering of different choices. The bias here preserves ordering. That is, if info_gain(x_1) > info_gain(x_2) then E[ info_gain_MC(x_1)] > E[info_gain_MC(x_2)]. K=16 was a statement regarding variance – note that some variance isn’t much of an issue for us, after all we are intentionally ‘injecting’ noise at the Boltzmann sampling stage, in order to obtain diverse samples.

---

> ### Author Response · Authors · 2018-01-05
> **Regarding the use of dropout**
>
> The paper https://arxiv.org/abs/1711.02989 refers to variational Gaussian dropout. We use Bernoulli dropout, which is a theoretically-grounded way of obtaining uncertainty estimates in neural networks. This method has already been used to obtain uncertainty estimates in Bayesian neural networks in several previous works:
>
> https://arxiv.org/abs/1506.02142
> https://arxiv.org/abs/1512.05287
> https://arxiv.org/abs/1703.02910

---

> ### Author Response · Authors · 2018-01-05
> **Appropriateness of eq. (2) as a target**
>
> While the proposed target distribution is not uniform over $\mathcal{X}_+$, it has the following advantages:
>
> (a) It functions as an indicator of validity, giving zero probability mass to invalid sequences.
> (b) It can be combined with generative models trained on real-world data which do not generate uniform samples. The proposed method can then be used to eliminate, at each step during the sequence generation in such models, those next actions (characters) that will lead to invalid sequences, improving the validity of the sequences generated.
> (c) It is invariant to changes in the training data distribution (active learning strategy).
> (d) It can handle padded sequences with no extra effort.
> (d) It is numerically stable -- with the output at each step being in the range [0, 1], rather than perhaps the ratio of sequences with that prefix that can lead to valid sequences, which tends to 0 with increasing sequence length and has typical scale that varies with step t.
>
> We also considered and tested as target a distribution that would be uniform over all sequences if trained with data distributed uniformly from $\mathcal{X}$. We found however that: (a) this only held for fixed length sequences and was not appropriate for padded sequences; (b) the requirement of uniform data from $\mathcal{X}$ prevented us from using active learning or any already existing data and; (c) the resulting method suffered from severe numerical/optimisation issues.

---

### Official Review · AnonReviewer1 · 2017-11-25
**Novel Approach to Generate Discrete Structure using RL**

**Rating:** 7
**Confidence:** 3

**Review:**

Overall: Authors casted discrete structure generation as a planning task and they used Q-learning + RNNs to solve for an optimal policy to generate valid sequences. They used RNN for sequential state representation and Q-learning for encoding expected value of sub-actions across trajectory - constraining each step's action to valid subsequences that could reach a final sequence with positive reward (valid whole sequences).

Evaluation: The approach centers around fitting a Q function with an oracle that validates sub-sequences. The Q function is supported by a sequence model for state representation. Though the approach seems novel and well crafted, the experiments and results can't inform me which part of the modeling was critical to the results, e.g. was it the (1) LSTM, (2) Q-function fitting? Are there other simpler baseline approaches to compare against the proposed method? Was RL really necessary for the planning task? The lack of a baseline approach for comparison makes it hard to judge both results on Python Expressions and SMILES. The Python table gives me a sense that the active learning training data generation approach provides competitive validity scores with increased discrete space coverage. However the SMILES data set is a little mixed for active vs passive - authors should try to shed some light into that as well.

In conclusion, the approach seems novel and seem to fit well with the RL planning framework. But the lack of baseline results make it hard to judge significance of the work.

---

> ### Author Response · Authors · 2018-01-05
> **SMILES results mixed for active vs passive**
>
> The reason for not using our active learning strategy with SMILES is because while it does learn to discover strings that are technically valid, they are not chemically-realistic. This is why our initial active learning results in the SMILES domain were mixed. By instead augmenting an existing set of realistic molecules, we are able to more efficiently explore the space of realistic SMILES strings.
>
> In our new SMILES experiments, instead of the active learning we propose a data augmentation strategy which generates informative negative samples. Table 4 shows that this strategy allows us to outperform previous state-of-the-art results [Kusner et al. (2017)].

---

> ### Author Response · Authors · 2018-01-05
> **Regarding experiments/baseline approaches**
>
> We have updated and extended our SMILES experiments. We now provide comparisons of our work with a state-of-art context-free grammar based approach [Kusner et al. (2017)]. This more clearly demonstrates the significance of our contribution.

---

> ### Author Response · Authors · 2018-01-05
> **Clarifying most relevant contribution**
>
> Our main contribution is the formulation of the problem as learning a Q function. To learn this function, however, we need informative data. For Python strings, where no positive data is available, we propose an active learning strategy to learn efficiently. For SMILES, where existing positive data is available, we propose a data augmentation strategy which allows us to obtain informative negative samples. We chose to describe our Q function with a recurrent neural network (LSTM), but any other similar model (GRU) could have been used as well.

---

### Official Review · AnonReviewer3 · 2017-11-27
**An interesting paper about a truly relevant problem. The proposed model seems to work well for SMILES strings representing moleculs, but its general applicability is a bit unclear to me. Further, I am not fully convinced about the novelty of the approach taken.**

**Rating:** 6
**Confidence:** 4

**Review:**

SUMMARY:
This work is about learning the validity of a sequences in specific application domains like SMILES strings for chemical compounds. In particular, the main emphasis is on predicting if a prefix sequence could possibly be extended to a complete valid sequence. In other words, one tries to predict if there exists a valid suffix sequence, and based on these predictions, the goal is to train a generative model that always produces valid sequences.  In the proposed reinforcement learning setting, a neural network models the probability that a certain action (adding a symbol) will result in a valid full sequence. For training the network, a large set of (validity-)labelled sequences would be needed. To overcome this problem, the authors introduce an active learning strategy, where the information gain is re-expressed as the conditional mutual information between the the label y and the network weights w, and this mutual information is maximized in a greedy sequential manner.
EVALUATION:
CLARITY & NOVELTY: In principle, the paper is easy to read. Unfortunately, however, for the reader is is not easy to find out what the authors consider their most relevant contribution. Every single part of the model seems to be quite standard (basically a network that predicts the probability of a valid sequence and an information-gain based active learning strategy) - so is the specific application to SMILES strings what makes the difference here?   Or is is the specific greedy approximation to the mutual information criterion in the active learning part? Or is it the way how you augment the dataset? All these aspects might be interesting, but somehow I am missing a coherent picture.
SIGNIFICANCE: it is not entirely clear to me if the proposed "pruning" strategy for the completion of prefix sequences can indeed be generally applied to sequence modelling problems, because in more general domains it might be very difficult to come up with reasonable validity estimates for prefixes that are significantly shorter than the whole sequence. I am not so familiar with SMILES strings -- but could it be that the experimental success reported here is mainly a result of the very specific structure of valid SMILES strings?  But then, what can be learned for general sequence validation problems?

UPDATE: Honestly, outside the scope of SMILES strings, I still have some concerns regarding reasonable validity estimates for prefixes that are significantly shorter than the whole sequence...

---

> ### Author Response · Authors · 2018-01-05
> **Clarifying most relevant contribution**
>
> Our main contribution is the formulation of the problem as learning a Q function. To learn this function, however, we need informative data. For Python strings, where no positive data is available, we propose an active learning strategy to learn efficiently. For SMILES, where existing positive data is available, we propose a data augmentation strategy which allows us to obtain informative negative samples. We chose to describe our Q function with a recurrent neural network (LSTM), but any other similar model (GRU) could have been used as well.
>
> To further demonstrate the the importance of our contribution, we’ve updated the SMILES experiments to include a comparison with previous work on validity of samples from VAE prior – a challenging domain where benchmarks exist. Here, our model sets the new state-of-the-art.

---

> ### Author Response · Authors · 2018-01-05
> **Significance and generalizability**
>
> Solving the validity learning problem in arbitrary domains can at worst be highly intractable. We believe practical solutions are only available when the validity rules are simple enough to be learned from data. Our approach is able to learn validity models in two very different domains, Python expressions and SMILES strings, demonstrating its capacity for generalization.
>
> The proposed active learning method is expected to be more beneficial in domains where shorter sequences demonstrate rules of validity that also apply to longer strings — that is, the nature of the governing validity rules does not change a great deal as sequences get longer.

---

> ### Author Response · Authors · 2018-01-05
> **Is the specific application to SMILES strings what makes the difference?**
>
> The proposed approach is applicable to any sequence validity problem in which our Q function is learnable from data. We considered the problems of learning the validity of python expressions and SMILES sequences because these are relatively simple problems that are also useful in practice and challenging for existing methods.

---

### Decision · Program_Chairs · 2018-01-29
**ICLR 2018 Conference Acceptance Decision**

**Decision:**

Accept (Poster)

**Comment:**

Viewing the problem of determining the validity of high-dimensional discrete sequences as a sequential decision problem, the authors propose learning a Q function that indicates whether the current sequence prefix can lead to a valid sequence. The paper is fairly well written and contains several interesting ideas. The experimental results appear promising but would be considerably more informative if more baselines were included. In particular, it would be good to compare the proposed approach (both conceptually and empirically) to learning a generative model of sequences. Also, given that your method is based on learning a Q function, you need to explain its exact relationship to classic Q-learning, which would also make for a good baseline.